# Symbolic Planning and Code Generation for Grounded Dialogue

**Justin T. Chiu**
Cornell Tech
jtc257@cornell.edu

**Wenting Zhao**
Cornell University
wz346@cornell.edu

**Derek Chen**
Columbia University
dc3761@columbia.edu

**Saujas Vaduguru**
Carnegie Mellon University
svadugur@andrew.cmu.edu

**Alexander M. Rush**
Cornell Tech
arush@cornell.edu

**Daniel Fried**
Carnegie Mellon University
dfried@cs.cmu.edu

## Abstract

Large language models (LLMs) excel at processing and generating both text and code. However, LLMs have had limited applicability in grounded task-oriented dialogue as they are difficult to steer toward task objectives and fail to handle novel grounding. We present a modular and interpretable grounded dialogue system that addresses these shortcomings by composing LLMs with a symbolic planner and grounded code execution. Our system consists of a reader and planner: the reader leverages an LLM to convert partner utterances into executable code, calling functions that perform grounding. The translated code's output is stored to track dialogue state, while a symbolic planner determines the next appropriate response. We evaluate our system's performance on the demanding ONECOMMON dialogue task, involving collaborative reference resolution on abstract images of scattered dots. Our system substantially outperforms the previous state-of-the-art, including improving task success in human evaluations from 56% to 69% in the most challenging setting.

## 1 Introduction

Success in grounded task-oriented dialogue requires intentional communication guided by strategic planning (Cohen and Perrault, 1979; Traum, 1994; Walker et al., 2004; Rieser and Lemon, 2009; Bakhtin et al., 2022, *inter alia*). Dialogue agents must read partner utterances, update their beliefs, then make a plan that furthers their goal. These plans must take into account both dialogue history and grounding, such as in an image. In end-to-end systems based solely on large language models (LLMs), this process is implicit and therefore difficult to control, requiring extra supervision (Christiano et al., 2023) or expensive search (Lu et al., 2022) to improve. While recent work has taken steps to rectify implicit reasoning via planning in language space, where intermediate steps are generated by an LLM (Wei et al., 2022), there is no

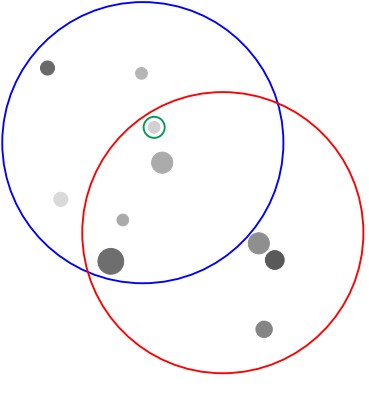

| | |
|---|---|
| SPC: | Do you see a pair of dots, where the top left dot is medium-sized and grey and the bottom right dot is medium-sized and dark |
| Human: | No |
| SPC: | Do you see a pair of dots, where the top dot is small-sized and light and the bottom dot is medium-sized and grey |
| Human: | Yes I see them. Is there a small grey dot above the small light dot? |
| SPC: | No. Is there a large size and dark color dot to the left and below those? |
| Human: | Yes and there is a small grey dot below them as well for me. |
| SPC: | Yes. Let's select the small size and light color one. |
| SPC: | <select> |
| Human: | <select> |

Figure 1: An example grounded dialogue from ONECOMMON. Our dialogue agent, SPC, and a human partner have different but overlapping circular views of a shared board. The agent and partner must collaborate through dialogue in order to find and select a shared dot. ONECOMMON demands careful, grounded reasoning.

guarantee that these approaches result in plans that further task progress. Additionally, planning in language space is expensive, requiring inference in an LLM (Yarats and Lewis, 2017; Guez et al., 2012).

Rather than implicit or heuristic reasoning, we are interested in explicit reasoning and planning over symbolic actions. Symbolic actions are controllable by construction, allowing system designers to easily build in task-specific knowledge (He et al., 2018; Bakhtin et al., 2022). This controlla-

bility is crucial for obtaining task-specific success using general tools, even with LLMs.

We provide an example from ONECOMMON, a particularly challenging grounded dialogue game (Udagawa and Aizawa, 2019). The goal of ONECOMMON is to, through dialogue, identify one dot in common with your partner, who has an overlapping but different view of an underlying set of dots, illustrated in Figure 1. The challenge in ONECOMMON is grounding the contextual spatial relationships described in language to dots.

Recent work has utilized code-generation for grounded language understanding (Dídac et al., 2023). In particular, they translate natural language questions to code as an intermediate representation, then execute that code to obtain an answer. Code has a couple appealing properties as an intermediate representation: First, modern language models are trained on a mixture of code and natural language, affording them the capability of, with some accuracy, translating between the two (Chen et al., 2021). Second, code acts as a compositional knowledge representation. This allows code-generation systems to perform grounded compositional reasoning, provided a library of Python functions that perform grounding (Liang et al., 2022).

We present a system, Symbolic Planning and Code-generation (SPC), that *reads* by translating partner utterances into code and *plans* based on symbolic reasoning over what to say next. Code as a compositional knowledge representation closely mirrors the compositional nature of utterances, which are composed of grounded parts. SPC plans by optimizing expected information gain, which has been shown to be effective at building a key aspect of collaborative dialogue: common ground (Yu et al., 2019; White et al., 2021; Chiu et al., 2022). Symbolic planning allows SPC to explicitly and efficiently optimize for task success while taking advantage of task-specific properties.

We evaluate our SPC system on the most challenging subset of the ONECOMMON task, comparing our system to the previous state-of-the-art supervised system for the task (Fried et al., 2021). In both evaluations with human partners and automated self-play evaluations, we find that our approach substantially outperforms the previous state-of-the-art in task accuracy, improving from 56% to 69% accuracy, and obtains comparable task accuracy to human-human pairs on average.

## 2 Related Work

Prior work on collaborative reference games focuses on building common ground (He et al., 2017; Haber et al., 2019; Khani et al., 2018). Prior work by Fried et al. (2021) implements an approximation of pragmatic reasoning on ONECOMMON, but plans in language space and utilizes supervised models for mapping language to symbols. Khani et al. (2018) plan in symbolic space, but without natural language. We plan in symbolic space and map from language to symbols via code generation.

Dialogue systems have a long history of reasoning with symbolic actions. When available, symbolic actions have been found to improve the performance of dialogue systems, especially in the setting of grounded dialogue (Winograd, 1971; Young, 2006; He et al., 2018; Andreas et al., 2020; Bakhtin et al., 2022). The closest work to ours is CICERO, which utilizes symbolic planning in a system for DIPLOMACY, a dialogue and strategy game that requires negotiation and coordination between players (Bakhtin et al., 2022). CICERO requires a supervised dataset to train their system. We use code LLMs which require minimal supervision beyond constructing a small perceptual grounding API.

Planning in dialogue systems has recently eschewed symbolic actions in favor of planning directly in text, where systems either perform rollouts, tree-search, or other forms of intermediate reasoning in language. This allows system designers to avoid manually defining symbolic actions (Yarats and Lewis, 2017; Jang et al., 2020; Gandhi et al., 2023). However, the accuracy of language-space planners is still low in many settings (Fried et al., 2021; Valmeekam et al., 2023). We focus on symbolic planning, where planning is defined in a space that ensures accuracy and controllability.

With the recent progress in large language modeling, code generation for modular grounded systems has quickly gained interest. Grounded code generation systems do not require task-specific training data, making them cheap to apply. A body of work utilizes a large language model for instruction following by generating Python code that makes calls to lower-level perception libraries (Liang et al., 2022; Dídac et al., 2023; Gupta and Kembhavi, 2022; Gao et al., 2023). This extends prior work on executable semantic parsing (Liang, 2016; Johnson et al., 2017; Cheng et al., 2018) with large language models. Concurrent work has also utilized code-generation to interpret language,

integrated with symbolic reasoning (Wong et al., 2023). We apply these advances to the setting of grounded task-oriented dialogue, where code generation grounds language to symbolic actions for use in explicit planning.

## 3   Overview: Reference Games

Collaborative reference games pair an agent and a partner in order to build common ground through natural language dialogue (Haber et al., 2019; Khani et al., 2018; He et al., 2017; Udagawa and Aizawa, 2019). Mirroring realistic scenarios, many reference games are also partially observable, where the agent and partner have different perspectives, and so they must resolve ambiguity.

ONECOMMON (Udagawa and Aizawa, 2019), as shown in Figure 1, is a reference game that exemplifies two challenges: grounding and planning. In ONECOMMON, the agent and partner see different but overlapping views of a set of dots, and the goal is to find and select one dot common to both players' views. Grounding in ONECOMMON is particularly difficult due to the dot-based visual context, which requires abstract spatial reasoning. Planning is complicated by the partial observability caused by differing perspectives, which require agents to use complex referring expressions in order to avoid ambiguity.[1] We focus on ONECOMMON due to its simplicity and difficulty.

Our approach to grounded reference games separates symbolic reasoning from language, allowing explicit steering. Our system, Symbolic Planning and Code-generation (SPC), breaks down a turn into three procedures: reading, planning, and writing. Reading and writing convert from language to symbols and vice versa, while planning reasons in purely symbolic space.

The agent maintains a belief distribution over possible worlds, $z$, representing task-specific unknowns. The goal of dialogue is to gain information about $z$ until the agent is confident enough to end the game. At each turn, the agent **reads** the partner's utterance $u$, converting it into a symbolic action, $p(x|u)$. This symbolic action potentially builds upon the action $x'$ of a previous utterance,

---

[1]The contexts in ONECOMMON were constructed to make referring expressions challenging and context-dependent. For example, if the agent sees only light dots, a relatively 'dark' dot for the agent may not be considered dark at all by the partner. ONECOMMON is an ideal testbed for pragmatic methods that reason about contextual meaning. While our approach does not address pragmatics, we hope future work will.

$u'$. The agent then **plans** in symbolic space. The system uses reasoning to update its belief state, $p(z|u) = \sum_x p(z|x)p(x|u)$, then produces a response $y^*$ of what to say next, which it describes in language to the partner. There is additionally a templated write module for generating a response from $y^*$ described in Appendix C.

In ONECOMMON, given a set of dots $\mathcal{D}$, the state $z \in \{0, 1\}^{|\mathcal{D}|}$ represents which dots the agent believes are contained (1) and not contained (0) in the partner's view, illustrated in Figure 3. We call a set of dots a *configuration*. The action representation of partner, $x$ and $x'$, and agent utterances, $y^*$, alike is also a configuration in $\{0, 1\}^{|\mathcal{D}|}$, as well as any answers or confirmations to previous questions.

## 4   Reading: From Language to Symbols

Reading in SPC requires interpreting utterances to a grounded symbolic action, which in turn facilitates the planning stage. Consider the following exchange:

> Agent:    *Do you see a triangle of dark dots?*
> Partner:  *Yes, is there a small grey one below it?*

Reading has several challenges. First, reading requires grounding utterances in context, e.g. the shapes and relations. Second, utterances are compositional. For example, the partner utterance builds on top of the previous utterance through coreference. Finally, a reading system must act quickly, as real-time dialogue systems require reasonable response times.

### 4.1   Code Generation

In SPC, reading is implemented as code generation. Given a dialogue, we generate Python code[2] which is then used as a meaning function to produce a distribution over all valid interpretations of the utterance's symbolic action (Figure 2). The code calls perceptual library functions with grounded semantics, drawn from a task-specific API. This perceptual library allows the system to both ground elements of the utterance and compositionally build upon previous utterances. Consider the following abbreviated example, based on ONECOMMON:

---

[2]We target Python as our code representation since it is well-understood by large language models. However, in principle, our system could target other languages such as Prolog or SQL.

```
from perceptual_library import is_small, ...
dot1, dot2, dot3, ... = get_dots()
```

Agent:     *Do you see a triangle of dark dots?*

```
agent_configs = set([
  Config(dot1, dot2, dot3),
  Config(dot3, dot4, dot1)
])
```

Partner:    *Yes, is there a small grey one below it?*

```
def turn(prev_configs):
  configs = set()
  for prev_config in prev_configs:
    for dot in single_dots(exclude=prev_config):
      if (
        is_small(dot)
        and is_grey(dot)
        and is_below(dot, prev_config)
      ):
        configs.add(Config(dot, prev_config))
  return configs
partner_configs_x = turn(agent_configs)
```

The code in the meaning function is imperative, but represents a set of declarative constraints representing $p(x|u)$.[3] The meaning function for the partner turn, turn(prev_configs), takes as input the distribution over symbolic actions of a previous turn, $p(x')$, and yields a set of possible interpretations of the current turn, $p(x|u) = \sum_{x'} p(x|u, x')p(x')$.[4] Because utterances can have multiple valid interpretations due to ambiguity, prev_configs represents a distribution.[5]

Within turn, we consider all valid configurations while marginalizing over $x'$, i.e. interpretations in prev_configs. For each interpretation, each dot is considered. If the new dot satisfies the semantics of the utterance, checked step-by-step via grounded perceptual library functions such as is_small(d), then it is a valid interpretation of the current utterance and is used to create a new Config.

The perceptual library functions are drawn from a manually-defined library. For ONECOMMON, we define these functions using domain-specific knowledge:

```
def is_small(d): return d.size < -0.3
```

The perceptual library for ONECOMMON can be

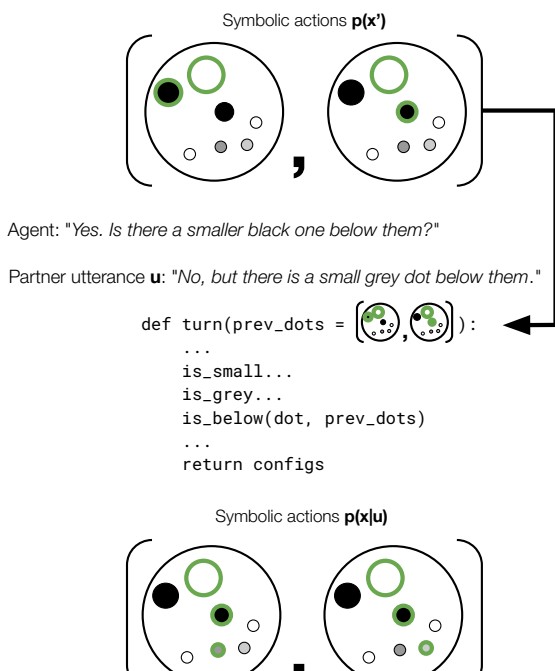

Partner utterance: "*Is there a big light dot next to a big dark one?*"

Symbolic actions **p(x')**

Agent: "*Yes. Is there a smaller black one below them?*"

Partner utterance **u**: "*No, but there is a small grey dot below them.*"

```
def turn(prev_dots =            ):
    ...
    is_small...
    is_grey...
    is_below(dot, prev_dots)
    ...
    return configs
```

Symbolic actions **p(x|u)**

Figure 2: Overview of Reading. The generated meaning function for utterance $u$ takes the previous symbolic action distribution $p(x')$ from a prior turn and yields the interpretations $p(x|u)$, using code as a compositional representation (Section 4).

found here.

## 4.2 Prompting

Reading is implemented with large language model (LLM) code generation. While LLMs can generate accurate code, full code specifications (Section 4.1) are lengthy and therefore too slow to generate for real-time use. We break down code generation into four steps, where some steps do not require any calls to an LLM. Decreasing the number of output tokens guarantees a speedup, assuming consistent latency. See the code for details on the code LLM and prompts we use.[6]

*Dialogue Act:* Classify partner utterances as one of three dialogue acts: Start a NEW line of questioning, ask a FOLLOW-UP question, END the dialogue.

*Reference:* Predict which previous turn $x'$ the utterance is following up on, if any:

Agent:     *Do you see a triangle?*

Partner:    *Yes, is there a small grey dot below it?*

```
dialogue act: follow-up
refer: turn 1
```

---

[3]In ONECOMMON, the distribution over symbolic actions $p(x|u)$ is represented as represented as a categorical distribution over configurations with probabilities based on the size of the circumcircle.

[4]The symbolic action of a previous turn $x'$ may also depend on other previous utterances $u'$. For simplicity, we omit that in the notation.

[5]SPC is able to intentionally produce ambiguous descriptions if that improves task success, as illustrated in this example.

[6]We release the code here.

The system grounds the dots mentioned in the previous turn: `agent_configs`, which is stored by the system. This allows referring to other turns besides the previous.

*Constraint Generation:* Predict the new dots mentioned in the partner utterance alongside code fragments that express the semantics, without the boilerplate code, in the example above:

> Partner:   *Yes, is there a small grey one below it?*
>
> ```
> 1 new dot
> is_small(dot)
> is_grey(dot)
> is_below(dot, prev_dots)
> ```

*Compose:* Finally, we utilize a template to compose all of this information back into the full code representation for execution.

## 5 Planning: From Symbols to Responses

To perform well in collaborative reference games, it is essential to build common ground quickly and accurately by carefully reasoning about what information has been gathered so far, as well as what to say next. SPC addresses these desiderata by planning in symbolic space, over the symbolic actions produced by reading.

We have two challenges: First, to incorporate the new information from the partner's utterance while accounting for task-specific grounding as well as dialogue history. Second, given this new information, the system must decide either to end the game or how to improve the probability of success.

Planning requires us to model the actions of the partner given the shared state. To do this we need task specific models of our partner, $p(x \mid z)$, and our partner's reponse to us, $p(x|z, y)$. In ONECOM-MON, we model both of these by a heuristic function considering set overlap and dot proximity, described in Appendix D.

### 5.1 Belief update

Starting from a prior over the previous belief $p(z)$, we incorporate probabilistic evidence from the utterance $p(x|u)$. This requires marginalizing over all valid symbolic actions $x$ from the reading step. In practice, $p(x|u)$ is sparse, and symbols $x$ with non-zero support are very similar. We therefore approximate this marginalization with a point esti-

mate:

$$
\begin{aligned}
p(z|u) &= \sum_x p(z|x)p(x|u) \\
&= \sum_x \frac{p(x|z)p(z)}{p(x)}p(x|u) \\
&\approx \sum_x \frac{p(x|z)p(z)}{p(x)}1(x = x^*) \\
&\propto p(x^* \mid z)p(z),
\end{aligned}
\tag{1}
$$

where $x^* = \operatorname{argmax}_x p(x|u)$.

We give an example of this process in Figure 3. In this case, a 'big light dot next to a big dark one' could have two valid interpretations, the big light dot and the black dot to the left, or the other black dot to the right. We approximate this distribution with the most likely interpretation $x^*$. In ONECOM-MON, we use the most compact[7] as $x^*$, yielding the black dot on the left. The belief state is then updated to $p(z|u)$, shown in Figure 3 (center).

### 5.2 Planning

Given the updated belief, SPC then plans its next action. The challenge here is to ensure task success, e.g. finding one dot in common. This requires both exploring by building common ground, then exploiting that knowledge to win the game.

We formalize exploration as the expected information gain, a quantity that codifies how much the agent can expect to learn about possible worlds $z$ after taking an action (Lindley, 1956). That action then elicits a response from the partner, providing information about the uncertain world state. For example, if the agent has asked about a set of dots and already received a 'no', then asking further questions about those dots would not reduce uncertainty.

Formally, we optimize

$$
y^* = \operatorname*{argmax}_y H[z|u] - \mathbb{E}_{x_y|y}\left[H[z \mid u, y, x_y]\right],
\tag{2}
$$

where $H[z|u]$ is the entropy of the current belief[8] and $H[z \mid u, y, x_y]$ the entropy of the posterior distribution. This second term is the key part of the objective. Assuming that we take action $y$, the expectation considers all hypothetical future

---

[7]We define the compactness of a configuration as the radius of the circumcircle. An ideal approximation would take into account more context, such as the relative sizes.

[8]The belief entropy $H[z|u]$ in the definition of information gain is constant with respect to the plan $x$, and can be dropped from the objective.

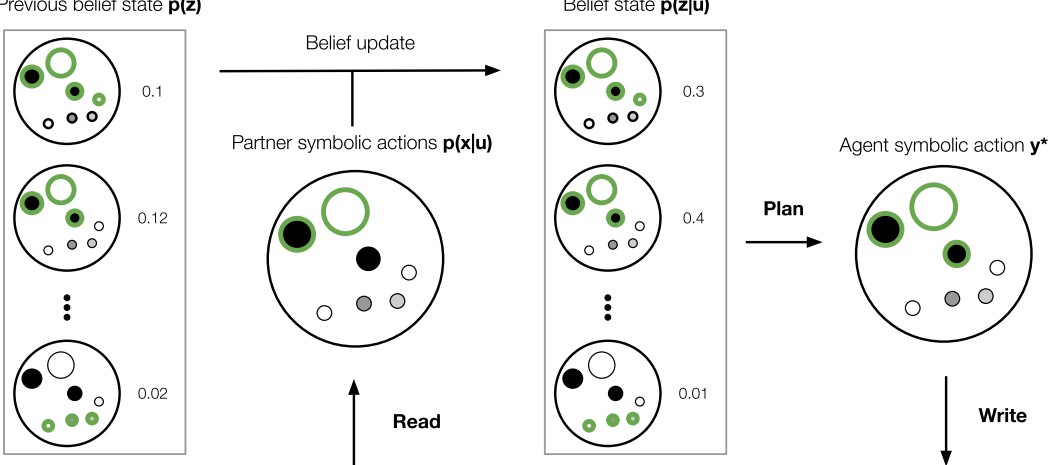

Previous belief state **p(z)**     Belief update     Belief state **p(z|u)**

0.1    0.12    0.02

Partner symbolic actions **p(x|u)**

0.3    0.4    0.01

Agent symbolic action **y\***

**Plan**    **Read**    **Write**

Partner utterance **u**: "*Is there a big light dot next to a big dark one?*"    Agent: "*Yes. Is there a smaller black one below them?*"

Figure 3: Overview of Planning. Partner utterances are interpreted by a meaning function generated by a code LLM (read), producing a distribution over valid symbolic interpretations, $p(x|u)$. This is used to symbolically update the belief state, $p(z|u)$, increasing the probability of worlds (shared dots) that are consistent with $x$. This belief state is used to symbolically plan the agent's next utterance, $y^*$, by optimizing the expected information gain, which is described to the partner (write).

partner responses $x_y$. We are penalized if after seeing these responses, we are still uncertain about the common ground $z$. This objective therefore encourages actions that reduce uncertainty. [9]

SPC chooses to exploit and end the game with the following heuristic: If the system is confident in success, i.e. the probability of task success is greater than hyperparameter $\theta$ (set to $0.8$), SPC ends the game.

## 6 Experimental Setup

We conduct two evaluations of SPC on the ONECOMMON task. We compare to the state-of-the-art baseline system of Fried et al. (2021), which we refer to as Imitate. Imitate is a pipelined system, where each part is fully supervised. Imitate uses a neural representation of dialogue history in combination with a neural-CRF reference resolution module to understand grounded language. In order to generate, Imitate relies on a pragmatic planning procedure, which plans in a mixture of symbolic and language space, prioritizing descriptions of dots that are easily understood.

We first perform human evaluation, evaluating the task success of systems when paired with human partners. This setting is challenging, requiring the system to handle both the linguistically diverse utterances and a range of strategies of human partners. We recruit 19 workers from Amazon's Mechanical Turk to play with one of three partners: SPC, the most successful past system for the task (Fried et al., 2021), or another human. We pay $15 per hour, with $1.00 per game at an average of 4 minutes per game. We additionally give a bonus of $0.15 for every game. We use 100 visual contexts from the most difficult[10] partition of ONECOMMON. We pay workers $1.00 per game, with a $0.15 bonus if they win. We collect 287 completed dialogues in total, where both players selected a dot.

We secondarily evaluate systems in self-play, where systems are paired with a copy of themselves. This isolates strategic efficiency by ensuring the agent's partner has the same skill as the agent. The 200 games share the same contexts across systems.

We include an additional system in self-play, GPT4 2-shot[11], which gets two full human dialogues as examples. Each human dialogue example starts with a description of the context the agent sees. The full prompts can be viewed here.

**Parameterization** For code generation, during the reading phase we use GPT-4[12] (OpenAI, 2023).

---

[9]The distribution $p(x_y|y) = \sum_z p(x_y|y,z)p(z)$ also uses the partner response model $p(x_y|y,z)$.

[10]The number of shared dots is four.

[11]We do not include GPT4 2-shot in human evaluation, as its self-play evaluation is very poor.

[12]Specifically `gpt-4-0613`.

| Agent | Success | Turns | Games |
|-------|---------|-------|-------|
| SPC | 68.8% | 7.77 | 96 |
| Imitate | 55.6% | 6.61 | 117 |
| Human | 67.6% | 5.03 | 74 |
| Human[†] | 65.8% | 4.97 | 2,189 |

Table 1: The average success rate, average number of turns, and total number of games between agents and human partners on the hardest setting of ONECOMMON, with 4 shared dots. [†] indicates statistics from the ONECOMMON dataset (Udagawa and Aizawa, 2019).

The symbolic actions in ONECOMMON consist of sets of dots and confirmations, while the belief over symbolic states, $p(z)$, captures which dot configurations are shared and is designed to account for dot proximity. Further details on the prior are given in Appendix D. The symbolic partner models, $p(x \mid z)$ and $p(x \mid y, z)$, are drawn from Chiu et al. (2022), and incorporate a similar bias based on dot proximity.

## 7 Results

**Human evaluation**  In human evaluation, SPC obtains substantially higher task accuracy than the baseline model of Fried et al. (2021), and is comparable to human performance on average. This demonstrates that the combination of symbolic information-gain planning and code-generation in SPC is more effective than the baseline's language-space planning objective and supervised reference resolution.

We see a more nuanced story when conducting a skill-based analysis of the human evaluation results, presented in Figure 4. A worker's skill is given by their average success rate with other human partners. The x-axis of the graph, the minimum success rate, increasingly filters workers from left to right: the left side of the graph shows all workers, while the far right shows only those workers who won nearly all of their human-human games. Skilled human partners have a higher success rate with other humans, as opposed to when partnered with SPC. Additionally, the success rate of SPC improves with human skill, while the success rate of human partners with the baseline system, Imitate, remains relatively constant across skill levels, implying that SPC is more responsive than the baseline to strategies used by humans.

SPC also takes more turns on average than both

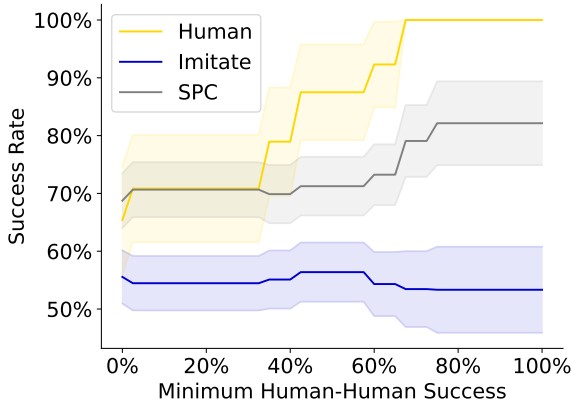

Figure 4: Success rate of the different agent types with human partners, with progressive filtering of human partners by their success rate along the x-axis. Shaded regions give standard errors.

| Agent | Avg $|u|$ | Median $|u|$ |
|-------|-----------|--------------|
| SPC | 6.95 | 4 |
| Imitate | 9.62 | 8 |
| Human | 15.06 | 14 |

Table 2: The average and median number of words per utterance by human partners for different agent types in human evaluation.

the baseline and human-human games. We hypothesize that this difference is caused by shorter human partner responses to the system, and therefore less information shared by the human partner. In Table 2, we confirm that the average and median number of words per human utterance are significantly lower for humans partnered with SPC than any other agent type.

**Self-play**  Similarly to human evaluation, SPC outperforms the baseline Imitate system in self-play as shown in Table 3. Compared to the baseline, SPC takes more turns on average, but has a higher success rate. We attribute both the longer games and higher success to symbolic planning, which ensures conservative playing. Interestingly, SPC self-play takes fewer turns on average than SPC-human pairings. We hypothesize that this is due to both copies of SPC communicating a consistent amount of information every turn. This also highlights the importance of human evaluation, which evaluates with a large population of partners.

We also find that GPT4 2-shot performs poorly in self-play. We attribute this to overly-agreeable responses, where the agents choose a dot without

| Agent | Success | Avg # turns |
|---|---|---|
| SPC | 84.0% | 4.83 |
| Imitate | 63.5% | 3.31 |
| GPT4 2-shot | 19.0% | 9.26 |
| Human[†] | 65.8% | 4.97 |

Table 3: The success rate of different agents in 200 self-play games on the hardest setting of ONECOMMON, with 4 shared dots. A higher success rate is better. The human performance is from the ONECOMMON dataset (Udagawa and Aizawa, 2019).

| Prompt style | Acc | Time (s) | Len |
|---|---|---|---|
| SPC | 86.7% | 5 | 36 |
| Full | 84.0% | 18 | 176 |

Table 4: The average accuracy, speed, and output length (number of tokens) for the sequential and full code generation methods in our benchmark reading task.

thorough verification or reasoning. This occurs despite the much longer dialogues, in comparison to all other agent types.

## 8 Analysis

**Qualitative analysis** We present a qualitative example of a dialogue between SPC and a human partner in Figure 5. This long dialogue illustrates the benefits of explicit belief updates and planning: The conversation starts off with many rejections, which the agent uses to deduce the shared configurations. Eventually, a common set of dots is found and the game ends in a success. Without explicit planning, it would have been unlikely for SPC to have succeeded at the end of the conversation.

**Reading speed analysis** We perform a speed ablation of the code-generation prompt in SPC. SPC uses a sequence of steps for reading, involving dialogue act classification, code fragment generation, and composing the full code representation based on the output of these steps. We compare this to a prompt that generates the full meaning function.

We evaluate both of these prompts in a reading task, where the goal is to read utterances generated by SPC and recover the underlying plans, measured by accuracy. In Table 4, we see that both styles of prompts have similar similar accuracy, but the sequential, decomposed approach is much faster due to shorter outputs.

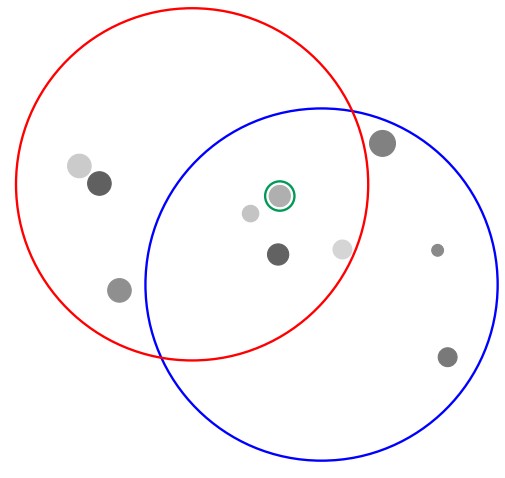

| | |
|---|---|
| SPC: | Do you see a pair of dots, where the bottom dot is medium-sized and dark and the top dot is small-sized and light |
| Human: | No. Do you see a pair of medium sized dots, close together, one is dark grey the other light grey. The light grey one is slightly above and the left of the dark one. |
| SPC: | No. Do you see a pair of dots, where the left dot is medium-sized and dark and the right dot is medium-sized and light |
| Human: | No, do you see a lone medium sized grey dot? |
| SPC: | No. Do you see a pair of dots, where the left dot is medium-sized and light and the right dot is small-sized and grey |
| Human: | No. do you see a pair where the right one is medium and grey and the left one is smaller and lighter. The smaller one is slightly below the medium sized one. |
| SPC: | Yes. Is there a medium size and light color dot to the right and below those? |
| Human: | Yes |
| SPC: | Let's select the medium size and grey color one. |
| Human: | <select> |
| SPC: | <select> |

Figure 5: An example dialogue between SPC and a human partner.

## 9 Conclusion

We present Symbolic Planning and Code-generation (SPC), a method that approaches grounded task-oriented dialogue by separating symbolic reasoning from language. Our approach uses an LLM to generate executable code functions which represent the meaning of utterances, mapping from language to symbolic actions. We then symbolically track task progress using Bayesian reasoning, and explicitly plan the best actions to take next using an information gain objective. Despite using minimal supervision, beyond a task-specific API and few-shot examples, our approach substantially outperforms the state-of-the-art system for the ONECOMMON task in both human eval-

uations and automatic self-play evaluations.

Our work contrasts with recent work on planning in language space, which reasons implicitly (Wei et al., 2022; Yarats and Lewis, 2017; Guez et al., 2012). While less flexible than language reasoning, symbolic reasoning is both interpretable and modular. Future work should seek to improve the flexibility of symbolic reasoning (Wong et al., 2023).

Our work also represents a first step toward using general-purpose code as a representation for downstream dialogue and interaction tasks. Future work might explore code-based representations that afford more flexible interaction with people, e.g., representing a broader range of user actions, both linguistic and grounded, to construct broadly useful interactive systems. An ideal system would be able to synthesize these representations with minimal manual intervention.

## Acknowledgements

We thank Vivian Chen, Sanjiban Choudhury, Ge Gao, Omer Gul, Sedrick Keh, Woojeong Kim, Celine Lee, Jack Morris, Chenran Ning, Jacob Sharf, Alane Suhr, Nicholas Tomlin, Anne Wu, and Jiawei Zhou for discussions, game-playing, and feedback at various points in the process.

We also thank the Mechanical Turkers of Turker Nation for their efforts in game-playing.

JC is supported by NSF #2242302. AMR is supported by a Sloan Fellowship and NSF CAREER #2037519. SV and DF were supported by gifts from Google and from Autodesk Research.

## Limitations

Our system performs code execution given human input, opening our system to several risks, such as code injection and unauthorized access. Future work must strive to integrate code execution capabilities in a secure manner.

Our approach also requires the manual engineering of a domain-specific API, as well as a symbolic representation. Future work should seek to alleviate the amount of manual engineering in order to improve flexibility. We hope that methods in program synthesis can provide a solution.

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

## A  Prompt details

All prompts rely on few-shot prompting. Reformat has 5 few-shot examples, Classify has two dialogues with 15-turns total, Confirm has 9 examples, and Understand has two dialogues with 15 turns total. All examples were based loosely on 10 examples from the human-human games collected in OneCommon by Udagawa and Aizawa (2019). The same prompts were used in every context. The full prompts can be found here.

## B  Prompt ablation

We present an additional experiment on how the choice of few-shot examples affects the code constraint generation prompt, which is a key component of the reading step. The code constraints express the relationships between the mentioned dots, e.g. whether they form a triangle or their relative positions, shapes, and colors.

We take the first human utterance from 20 games in human evaluation and examine whether the parsed answer changes when the prompt examples are changed. The 15 examples in the constraint generation prompt were labeled by hand. Since we cannot sample another 15 examples, we instead sub-sample 5 random examples out of 15 for a 5-shot prompt. We report the average agreement between 5-shot prompts and the original 15-shot prompt across 5 trials: 99%, with a standard deviation of 2%. This implies the constraint generation prompt is not sensitive to prompt example choice at the 5-shot level and prompts could be further optimized.

We perform the same experiment with 5 trials of 1-shot prompts and see an average agreement rate of 34% with a standard deviation of 42%. This implies that given a single example, the prompt example matters.

We also find that a zero-shot prompt is unable to generate output in the correct format.

## C  Writing

We utilize three templates for writing, one for each dialogue act.

START: Do you see a pair of dots, where the {position} dot is {size}-sized and {color} and the {position} dot is {size}-sized and {color}?

FOLLOW-UP: Is there a {size} size and {color} color dot {position} those?

SELECT: Let's select the {size} size and {color} color one. <selection>

## D  Parameterization

We give the parameterization of the belief prior, $p(z)$ for ONECOMMON.

Our goal in designing the prior is to ensure that the closer dots are, the more likely they are to be of the same state: either all shared or not. This reflects the contiguity of ONECOMMON perspectives.

The prior is given by

$$p(z) \propto \exp(f(z)), \quad (3)$$

where $f(z)$ is given the sum of the edges of a minimum spanning tree for the dots in $z$. The weights of this spanning tree are determined by the rank of how close the dots are to each other. The edge between the nearest neighbor of a dot and the dot itself gets assigned a weight of 0, the 2nd nearest neighbor a weight of 1, and so on.

## E  Relation to prior work in semantic parsing and dialogue state tracking

Prior work in semantic parsing for dialogue state tracking, such as in SMCalFlow (Andreas et al., 2020), does not ground in a visual context and also requires strategic, collaborative planning due to OneCommon's symmetric roles. Agents must both give and request information strategically. This type of strategic reasoning is not explored in prior works in semantic parsing and dialogue state tracking. Our technical contribution is unifying grounded language understanding and strategic symbolic reasoning with code generation. In particular, the reading phase of SPC was designed for spatial reasoning in OneCommon.