# OpenReview forum: "Symbolic Planning and Code Generation for Grounded Dialogue"
_EMNLP/2023/Conference — EMNLP 2023 Main_

### Official Review · Reviewer_pMuZ · 2023-07-28

**Soundness:** 3

**Excitement:**

3: Ambivalent: It has merits (e.g., it reports state-of-the-art results, the idea is nice), but there are key weaknesses (e.g., it describes incremental work), and it can significantly benefit from another round of revision. However, I won't object to accepting it if my co-reviewers champion it.

**Missing References:**

There have been prior works on TOD systems grounded on some symbolic environment, either through code generation or implementing logic as a symbolic program. SMCalFlow (https://direct.mit.edu/tacl/article/doi/10.1162/tacl_a_00333/96470/Task-Oriented-Dialogue-as-Dataflow-Synthesis) generates “dataflow graphs” from natural language conversations, parsing semantic intents into executable programs. Shin et al., 2021 (https://arxiv.org/abs/2104.08768) use controlled decoding of a pretrained LLM on SMCalFlow. Zhao et al, 2022 (https://arxiv.org/abs/2212.09939) connect an LM to a symbolic dialog planner with arbitrary logic in order to perform enterprise-like TOD.

There is also much research being done in the field of “agentic LLMs”, which connect a LLM to a symbolic environment and have it act as its own agent, when predicting actions (https://arxiv.org/abs/2210.03629, https://arxiv.org/abs/2303.11366, inter alia).

**Paper Topic And Main Contributions:**

The authors demonstrate “Symbolic Planning and Code-Generation” (SPC), a system combining a prompted LLM (GPT-4) to perform as a task-oriented dialogue agent, and a symbolic planner that uses code generation. SPC is built for the OneCommon benchmark, a reference game between two players. Each player is given a separate view of a cluster of dots, and the objective of the game is for both players to communicate in natural language on the dots that they see, and to select a dot that is within both players’ viewpoints.

SPC is broken down into a pipelined set of behaviors. At each turn, the agent reads its partner’s utterance, and converts it into a probabilistic symbolic representation of the world state through code generation by a prompted GPT-4. This symbolic representation is then operated over by a partner model from Chiu et al. 2022 which updates SPC’s belief state about which dots are visible by both it and its partner. Outputs from the dialog planning model are converted into text utterances through templating.

SPC demonstrates SOTA results on OneCommon over prior baselines, during human evaluation when playing the OneCommon game.


**Questions For The Authors:**


1) Would a data release for SPC be possible? The outputs from the human evaluation of SPC may be valuable for future researchers on the OneCommon benchmark, especially for future comparison. This also includes the human-to-human evaluation.


**Reasons To Accept:**

This paper demonstrates an interesting task-oriented dialog (TOD) system that follows a neuro-symbolic architecture, in which a neural component (GPT-4) converts from natural language into a probabilistic symbolic representation, which is then operated over by a dialog planning model to control the behavior of the agent towards playing OneCommon successfully. This is a burgeoning field in using LLMs, in terms of grounding them in environments and ensuring that there is an interpretable and grounded representation of its belief state, leading to a SOTA result on OneCommon.

**Reasons To Reject:**

1. The specification for how SPC converts from its partner’s natural language utterances to a symbolic representation of the world is not very clear to me. This is described in Section 4.2 and Section 6 (L404-414), but there doesn’t appear to be a description of how GPT-4 is prompted to convert to the symbolic representation. From what I understand, the output of GPT-4 will resemble L290-292, which is then parsed by the “partner model” mentioned in L412. But there are still a lot of questions left unanswered — for example:

    a. What does GPT-4’s prompt look like?

    b. Are there few-shot examples in the prompt?

    c. Is there any regulation on how GPT-4’s output is kept in a parseable format, and what rules does the parser follow?

    d. What does the input to the partner model look like, and how is its output converted into a natural language utterance returned to the partner player?

    e. Can you provide a step-by-step description of how a partner’s utterance is read, parsed, and fed into the partner model, and how the partner model’s output is converted into generated text?

Much more detail needs to be provided in SPC’s methodology, and I would find it hard to reproduce the SPC system described in the paper.

2. The contribution that this paper makes over prior work done by Chiu et al., 2022 (https://openreview.net/pdf?id=PkHSHZLig5H) is unclear. This paper cites Chiu et al., 2022 and reuses its presented dialog planner, and indeed describes the same “information gain” approach when doing dialog planning on OneCommon. In particular, this paper’s Section 5.2 is identical to the theoretical decomposition described in Chiu et al.’s Section 3 (and there should be a citation here!).

    a. Am I correct in saying that the major contribution is embedding Chiu et al.’s dialog planner with a prompted GPT-4 that also feeds into the dialog planner with code generation? If so, this also needs to be made more explicit, and again more detail needs to be provided on the code generation aspect (as mentioned in (1)). If this is true, I'm not sure what the value of this work is with respect to prior works that also prompt large pretrained LLM's to interact with some symbolic task (e.g. https://arxiv.org/abs/2210.03629)

    b. What is the value of code generation from GPT-4 when interfacing with the partner model? Are there other ways to connect the partner model with some LM to do conversation?


**Reproducibility:**

3: Could reproduce the results with some difficulty. The settings of parameters are underspecified or subjectively determined; the training/evaluation data are not widely available.

**Reviewer Confidence:**

4: Quite sure. I tried to check the important points carefully. It's unlikely, though conceivable, that I missed something that should affect my ratings.

---

> ### Author Rebuttal · Authors · 2023-08-29
>
> Thank you for your thoughtful review. We are encouraged that you found the system interesting and the results positive. We will include all clarifications below in the next version:
>
> ## Prompts
> We apologize for not including examples of the prompts in the original submission. For reproducibility, the code and data for generating the prompts was included in the supplementary material. We provide anonymized links containing an example of each prompt and a pointer to the generating Jinja template in the supplementary material.
>
> Full prompt
> * Template: `oc-anon/oc/prompts/understand5.j2`
> * Link: https://pastebin.com/x8dxj4tz
>
> SPC prompts
>
> Reformat: Rewrite partner utterances to correct typos and formatting.
> * Template: `oc-anon/oc/prompts/reformat.j2`
> * Example: https://pastebin.com/uTeP8nZE
>
> Classify: Classify the dialogue act and any reference to previous turns.
> * Template: `oc-anon/oc/prompts/classify2.j2`
> * Example: https://pastebin.com/FFY8APRX
>
> Confirm: Classify whether the partner said yes or no to a previous question.
> * Template: `oc-anon/oc/prompts/confirm.j2`
> * Example: https://pastebin.com/YpTiTkSD
>
> Constraint generation: Generate the constraints.
> * Template: `oc-anon/oc/prompts/understandshort2.j2`
> * Example: https://pastebin.com/ube83h3n
>
> Execution template: Take the output of previous prompts and turn them into a program.
> * Template: `oc-anon/oc/prompts/executeshort2.j2`
> * Example: https://pastebin.com/JJvGAtyU
>
> ## Prompt constraints and parsing
> We do not impose any constraints on the GPT-4 output. If the generated code fails to execute or there is a malformed output, we ignore the partner utterance in that turn. Recent advances in constrained prompting should greatly improve such errors. For prompt parsing, we rely on the MiniChain library. The code can be found in `oc-anon/oc/prompt.py` in the supplementary material.
>
> ## Input to the partner model
> The partner model takes as input a query about whether a set of dots "can be seen", represented as a binary vector $\\{0,1\\}^7$ showing which dots are being referred to. The partner model’s output is a distribution over yes or no. The full code for the partner model can be found in `oc-anon/oc/belief/belief.py` in the supplementary material.
>
> ## Step-by-step description of SPC
> We expand upon the description in L186-200 below, and will include these clarifications in any future version of the paper, given extra space:
> * Reading: We map partner utterances to a set of dots they are asking about. We obtain the set of dots by prompting with either the full prompt (Section 4.1) or more efficient decomposed prompts (Section 4.2). We execute the generated code to return a set of dot referents. The parsing is performed via few-shot prompting, with the prompts given above. The code for reading can be found in `oc-anon/oc/agent2/reader.py` in the supplementary material.
> * Belief update: If SPC successfully resolves the set of dot referents, we update our belief (Section 5.1, Equation 1). The code for the belief update logic can be found in `oc-anon/oc/agent2/reader.py` in the supplementary material. The partner model code is in `oc-anon/oc/belief/belief.py`.
> * Planning: Given the updated belief, we consider all possible next actions (sets of dots to ask about) and reason about what would happen if we took those actions. We pick the action that would result in the most information gained about the shared dots (Section 5.2, equation 2). For dialogue games, computing the information gain from a particular action requires marginalizing over partner responses to said action. The code for planning can be found in `oc-anon/oc/agent2/planner.py` in the supplementary material.
> * Writing: We then describe the chosen action, a set of dots. We utilize one of three templates: a start template, follow-up template, or select template. The start template describes the set of dots, the follow-up template describes the new dots in relation to a previously mentioned set of dots, and the select template describes a single dot to select. We additionally add a “Yes” or “No” to the start of the response if the partner utterance contained a question that was successfully parsed and resolved. The templates refer to dots by their absolute size, color, and relative positions to other dots. The code for writing can be found in `oc-anon/oc/agent2/writer.py` in the supplementary material.
> ## Contributions over Chiu et al 2022
> Chiu et al 2022 [4] is a non-archival theoretical work that doe not have experiments on language selfplay or human evaluation. Our contribution, as you correctly describe, is combining the partner model-based symbolic planner with a semantic parser in order to perform grounded dialogue. This combination allows our system to generate and interpret language, allowing (1) self-play automatic evaluation that uses language (not just symbols) and (2) interacting successfully with human partners.
> ## Comparison to ReACT
> ReACT allows tool use, but still performs reasoning in language space. Compared to SPC, ReACT trades off correctness for flexibility. Language-space reasoning is not limited to pre-defined logical forms but can suffer from hallucinations. Planning in SPC is restricted to the manually defined symbolic space, but cannot hallucinate. We hypothesize that ReACT could potentially achieve similar performance to SPC by carefully describing OneCommon-specific heuristics and instructions in language, in combination with access to our grounded perceptual library. However, specifying the planning objective in symbolic space with SPC is more natural and allows us to take advantage of the theoretical partner model from Chiu et al 2022 [4].
> ## The value of code generation
> Code generation has two properties that make it appealing in OneCommon. First, it allows for the integration of grounded perceptual libraries with the symbolic planner (L79-L92). The grounded perceptual library is important because OneCommon requires careful spatial reasoning, and the symbolic planner provides an interface for designing task-specific constraints. Second, it provides a compositional knowledge representation that naturally describes declarative constraints (L64-78). Referring expressions in OneCommon are compositional, referring to groups of dots as well as individual features of particular dots. Code naturally composes individual feature constraints into group descriptions. Additionally, large language models are accurate at parsing these referring expressions to constraints in code.
> ## Alternative methods for connecting language and symbolic planning
> Rather than generating code as an intermediate meaning function to translate from language to symbols, it is possible to directly predict grounded referents. The reading component of the baseline "Imitate" system takes this approach. In preliminary experiments, we used the direct prediction reading model from "Imitate" as the reader in SPC and found that it cannot generalize to the simple template generations in SPC: 4/25 for direct prediction versus 22/25 with SPC’s reader. In particular, direct prediction with Imitate frequently mis-classifies referring expressions that are rare in the training data but easily understood by a human. This implies that the direct prediction reading model conflates the semantics of a referring expression with biases in training data. We turn to code generation with LLMs to decouple semantics from biases in training data.
> ## Missing references
> Thank you for the references, we will include these in the next version.
> ## Citation for information gain (Section 5.2, equation 2)
> Our citation for the planning objective is Lindley, 1956 [1] (L346). We will include two more references for this objective [2,3].
> ## Data release
> We will release the human evaluation data.
>
> [1] D. V. Lindley. 1956. On a Measure of the Information Provided by an Experiment. The Annals of Mathematical Statistics, 27(4):986 – 1005.
>
> [2] MacKay, David J. C. 1992. “Information-based Objective Functions for Active Data Selection”. Neural Computation 4 (4): 590–604.
>
> [3] Hennig, P., & Schuler, C.J. (2011). Entropy Search for Information-Efficient Global Optimization. J. Mach. Learn. Res., 13, 1809-1837.
>
> [4] Justin T. Chiu, Wenting Zhao, Daniel Fried, Alexander Rush. “Modeling Perspective-Dependent Ambiguity in Collaborative Dialogue.” NAACL 2022 Wordplay Workshop. Non-archival.

---

### Official Review · Reviewer_soD2 · 2023-08-02

**Soundness:** 4

**Excitement:**

3: Ambivalent: It has merits (e.g., it reports state-of-the-art results, the idea is nice), but there are key weaknesses (e.g., it describes incremental work), and it can significantly benefit from another round of revision. However, I won't object to accepting it if my co-reviewers champion it.

**Paper Topic And Main Contributions:**

The paper proposes a new method for the task of grounded dialogue, in which an agent needs to achieve an objective together with the partner based on some shared state that needs to be communicated.
The proposed approach leverages executable semantic parsing (using a subset of Python, generated by the LLM) to interpret the utterances received by the agent. The formal representation is executed deterministically to compute the most likely interpretation of the utterance, which is used to update a probability distribution over all possible shared states. The new probability distribution is then used to choose a configuration of the shared state to test that maximizes information gain, which is then asked to the partner by generating a question following a template. The game continue until one configuration has an estimated probability above a threshold.
The approach is evaluated on the OneCommon task, where agent and partner operate on a board containing shapes. Each agent and partner are assigned one subset of the shapes, and they need to identify at least one shape they have in common.

**Reasons To Accept:**

The main reason to accept is that the method is effective for the proposed task, achieving near human performance across all partners, and showing that it can adapt to advanced strategies adopted by humans who are highly successful at this task. Additionally, the method leverages the intrinsic ability of modern LLMs and does not require any new training.

**Reasons To Reject:**

The main weakness, which admittedly is also discussed as a limitation in the paper, is that the proposed method appears to only be realistically applicable to the OneCommon task, and not to more useful real-world grounding tasks.
In particular, the proposed method requires a hand-engineered library of primitive symbolic checks that can be efficiently computed in Python. It is unclear in which real-world settings this would be possible, for a number of reasons, including the inherent ambiguity and context dependence of concepts such as "small" or "large", and the fact that such a check might be non trivial to implement for, say, a picture of physical objects rather than an artificial collection of shapes.
Second, the proposed method requires all possible configurations of the shared state to be efficiently enumerable, as the generated code marginalizes over them (Section 4.1). In reality, this is probably intractable in all but the simplest artificial problems.

To be clear, as the method is effective on the stated task, this is a relatively minor weakness, but if the paper intends to show that the method is applicable to real-world tasks, then it needs an evaluation in a more realistic benchmark than just OneCommon.

A separate but related weakness of the paper is the choice to generate Python directly, as opposed to a dedicated DSL or a more common declarative PL such as SQL. Given the code generation is the core contribution of this paper, there should be more discussion of what exactly is the expressive power of the generated code, and how this contribution is different from previous work in semantic parsing and dialogue state tracking.

After rebuttal: Thanks for addressing all comments. I agree that extending the proposed approach to probabilistic programs rather than deterministic would greatly increase the applicability of the method. I also understand that such an extension could be out of scope for this work.
I would recommend including such discussion, as well as well discussion of prompts and choice of target language, in the final version.
Overall, I am satisfied that the concerns can be addressed in the final revision, and I updated my score.

**Reproducibility:**

2: Would be hard pressed to reproduce the results. The contribution depends on data that are simply not available outside the author's institution or consortium; not enough details are provided.

**Reviewer Confidence:**

3: Pretty sure, but there's a chance I missed something. Although I have a good feel for this area in general, I did not carefully check the paper's details, e.g., the math, experimental design, or novelty.

---

> ### Author Rebuttal · Authors · 2023-08-29
>
> Thank you for your thoughtful review. We value reviewer soD2's positive perspective on the convincing human evaluation results and the promising aspect of not needing model training. We will include all clarifications below in the next version:
>
> ## Prompts
> We apologize for not including examples of the prompts in the original submission. For reproducibility, the code and data for generating the prompts was included in the supplementary material. We provide anonymized links containing an example of each prompt and a pointer to the generating Jinja template in the supplementary material.
>
> Full prompt
> * Template: `oc-anon/oc/prompts/understand5.j2`
> * Link: https://pastebin.com/x8dxj4tz
>
> SPC prompts
>
> Reformat: Rewrite partner utterances to correct typos and formatting.
> * Template: `oc-anon/oc/prompts/reformat.j2`
> * Example: https://pastebin.com/uTeP8nZE
>
> Classify: Classify the dialogue act and any reference to previous turns.
> * Template: `oc-anon/oc/prompts/classify2.j2`
> * Example: https://pastebin.com/FFY8APRX
>
> Confirm: Classify whether the partner said yes or no to a previous question.
> * Template: `oc-anon/oc/prompts/confirm.j2`
> * Example: https://pastebin.com/YpTiTkSD
>
> Constraint generation: Generate the constraints.
> * Template: `oc-anon/oc/prompts/understandshort2.j2`
> * Example: https://pastebin.com/ube83h3n
>
> Execution template: Take the output of previous prompts and turn them into a program.
> * Template: `oc-anon/oc/prompts/executeshort2.j2`
> * Example: https://pastebin.com/JJvGAtyU
>
> ## Real-world applications require context-dependent meaning
> We agree that real-world applications require context-dependent meaning. OneCommon was designed as a testbed for context-dependent dialogue systems. In particular, size and color words in OneCommon refer to a wide and overlapping range of values that depend on context [1]. SPC does not take advantage of this directly, and we believe doing so would improve the accuracy of SPC. Future work could extend SPC with pragmatic meaning inference by moving from deterministic code as a meaning function to probabilistic programs, where size and color are represented as thresholds over a context-dependent random variable that must be inferred through interaction. Applying pragmatics through SPC would afford modularity, allowing the model to separately reason about size and color.
>
> ## Marginalization over actions is intractable in real-world applications
> We think there does exist important real-world tasks, such as image retrieval over a thousand candidate images, where exact marginalization over actions, images in the case of image retrieval, is indeed tractable. In more general settings, marginalization may be intractable. Prior work suggests that SPC like method could be generalized to such settings with approximate inference [2,3].
>
> ## Generalization to richer tasks
> We agree that future work should focus on applying the method to richer tasks, as well as decreasing the amount of manual effort required to craft a perception library. However, OneCommon is a challenging testbed for grounded dialogue as it is both simple and challenging. It emphasizes two important phenomena in grounded dialogue: spatial reasoning and context-dependent language understanding.
>
> ## Relation to prior work in semantic parsing and dialogue state tracking
> Our technical contribution is unifying grounded language understanding and strategic symbolic reasoning with code generation. In particular, the reading phase of SPC was designed for OneCommon, which requires spatial reasoning. Prior work in semantic parsing for dialogue state tracking, such as in SMCalFlow [4], does not ground in a visual context and also requires strategic, collaborative planning due to OneCommon’s symmetric roles. Agents must both give and request information strategically. This type of strategic reasoning is not explored in prior works in semantic parsing and dialogue state tracking.
>
> ## Python as a code representation
> We target Python as it is more flexible than domain specific languages and is well-understood by large language models. However, our decomposed prompts in particular do not rely on knowledge of Python, and could target other languages such as SQL.
>
> [1] Takuma Udagawa, Takato Yamazaki, and Akiko Aizawa. 2020. “A Linguistic Analysis of Visually Grounded Dialogues Based on Spatial Expressions.” In Findings of the Association for Computational Linguistics: EMNLP 2020, pages 750–765, Online. Association for Computational Linguistics.
>
> [2] Foster, Adam et al. “Deep Adaptive Design: Amortizing Sequential Bayesian Experimental Design.” International Conference on Machine Learning (2021).
>
> [3] Kevin Swersky, Yulia Rubanova, David Dohan, Kevin Murphy. “Amortized Bayesian Optimization over Discrete Spaces.” Proceedings of the 36th Conference on Uncertainty in Artificial Intelligence (UAI), PMLR 124:769-778, 2020.
>
> [4] Andreas, Jacob et al. “Task-Oriented Dialogue as Dataflow Synthesis.” Transactions of the Association for Computational Linguistics 8 (2020): 556-571.

---

### Official Review · Reviewer_mUba · 2023-08-04

**Soundness:** 4

**Excitement:**

4: Strong: This paper deepens the understanding of some phenomenon or lowers the barriers to an existing research direction.

**Paper Topic And Main Contributions:**

This paper introduces Symbolic Planning and Code-generation (SPC), a new method for grounded task-oriented dialogue. It is applied to the OneCommon task, which involves collaborative reference resolution on images of scattered dots, with each of the dialogue system and the human partner seeing a different subset of the dots (but overlapping). SPC utilizes an LLM (GPT-4 in this case) to understand the human partner's utterances and generate code that is subsequently used in the planning stage performed in symbolic space rather than language space. Templates are then used to go back from the symbolic space to language and generate a response. Human evaluation shows this method to greatly outperform the previous SOTA method. Self-play experiments show an even bigger performance difference between SPC and the previous SOTA method.

**Questions For The Authors:**

- A.) In Fig. 4, the mean success rate of SPC seems to be constantly above 70%, and going above 80% with skilled partners. How come the mean success rate reported in Table 1 is 68.8%?
- B.) Do you have a hypothesis for why the utterances of human partners were so much shorter when interacting with SPC than with Imitate? Does it have to do with the type of responses/questions each of these systems uses?

**Reasons To Accept:**

- The proposed method goes seamlessly between language space and symbolic space while utilizing an LLM. Performing the planning/reasoning explicitly in symbolic space, maximizing an information gain objective, is more effective than planning performed in language space as previous work has done when employing language models for this task.
- A lot of illustrative examples and diagrams that make the understanding of the task and the method easy.
- Reducing the code generated by the LLM down to just the set of the necessary predicates representing the essential information, in order to speed up the generation.
- Outperforming the previous SOTA approach by a large margin in a human evaluation (69% vs. 56% success rate).
- The performance of SPC actually steadily improves with the increasing skill of the human partner (which is the opposite of the previous SOTA method's behavior).

**Reasons To Reject:**

- There is no description whatsoever of how the LLM is prompted to produce the code fragments. Presumably, some examples are provided, as the LLM would probably not do a good job in a zero-shot setting. However, we don't know how many examples were provided in the prompt, how complex they were, and whether it was always the same set of examples or varied based on the context. The choice of examples in the prompt probably also affects the accuracy, but we don't know to what extent.
- The reading phase (~ 5 seconds on average) is still rather slow for real-time application.
- The median number of words per utterance by human partners interacting with SPC is merely a third of the median of the utterances in human-human interactions (4 vs. 14 words). While this does not necessarily mean anything bad, the large difference makes me wonder if the interaction with SPC feels very unnatural. Perhaps SPC is too controlling the way it asks questions (perhaps with most humans simply responding "yes" or "no") and could benefit from an improved response generation to allow for more user initiaitve.

**Reproducibility:**

3: Could reproduce the results with some difficulty. The settings of parameters are underspecified or subjectively determined; the training/evaluation data are not widely available.

**Reviewer Confidence:**

3: Pretty sure, but there's a chance I missed something. Although I have a good feel for this area in general, I did not carefully check the paper's details, e.g., the math, experimental design, or novelty.

**Typos Grammar Style And Presentation Improvements:**

- L212: extra "an"
- Footnote 1 on page 4: "represented as" repeated twice
- L486: "similar" repeated twice

---

> ### Author Rebuttal · Authors · 2023-08-29
>
> Thank you for your thoughtful review. We are encouraged that reviewer mUba found that the examples and diagrams were helpful and made the task and method easy to understand. We will include all clarifications below in the next version:
>
> ## Prompts
> We apologize for not including examples of the prompts in the original submission. For reproducibility, the code and data for generating the prompts was included in the supplementary material. We provide anonymized links containing an example of each prompt and a pointer to the generating Jinja template in the supplementary material.
>
> Full prompt
> * Template: `oc-anon/oc/prompts/understand5.j2`
> * Link: https://pastebin.com/x8dxj4tz
>
> SPC prompts
>
> Reformat: Rewrite partner utterances to correct typos and formatting.
> * Template: `oc-anon/oc/prompts/reformat.j2`
> * Example: https://pastebin.com/uTeP8nZE
>
> Classify: Classify the dialogue act and any reference to previous turns.
> * Template: `oc-anon/oc/prompts/classify2.j2`
> * Example: https://pastebin.com/FFY8APRX
>
> Confirm: Classify whether the partner said yes or no to a previous question.
> * Template: `oc-anon/oc/prompts/confirm.j2`
> * Example: https://pastebin.com/YpTiTkSD
>
> Constraint generation: Generate the constraints.
> * Template: `oc-anon/oc/prompts/understandshort2.j2`
> * Example: https://pastebin.com/ube83h3n
>
> Execution template: Take the output of previous prompts and turn them into a program.
> * Template: `oc-anon/oc/prompts/executeshort2.j2`
> * Example: https://pastebin.com/JJvGAtyU
>
> ## Prompt details
> All prompts rely on few-shot prompting. Reformat has 5 few-shot examples, Classify has two dialogues with 15-turns total, Confirm has 9 examples, and Understand has two dialogues with 15 turns total. All examples were based loosely on 10 examples from the human-human games collected in OneCommon by Udagawa and Aizawa. The same prompts were used in every context.
>
> ## How the choice of prompt examples affects accuracy
> We found that a zero-shot prompt was not able to generate output in the correct format, and therefore did not investigate further.
>
> We ran an additional experiment to evaluate how the choice of examples affects the code constraint generation prompt. We take the first human utterance from 20 games in human evaluation and examine whether the parsed answer changes when the prompt examples are changed. The 15 examples in the constraint generation prompt were labeled by hand. Since we cannot sample another 15 examples, we instead sub-sample 5 random examples out of 15. We report the average agreement between 5-shot prompts and the original 15-shot prompt across 5 trials: 99%, with a standard deviation of 2%. This implies the constraint generation prompt is not sensitive to prompt example choice at the 5-shot level and prompts could be further optimized.
>
> We perform the same experiment with 5 trials of 1-shot prompts, and see an average agreement rate of 34% with a standard deviation of 42%. This implies that given a single example, the prompt example matters.
>
> ## 5 seconds per response is slow
> The 5 second response time for SPC is comparable to the median typing time of 8.16 seconds per turn for humans in OneCommon, and could be further improved by running open models locally, such as StarCoder, for scenarios that require it.
>
> ## Difference in median length of human responses with human-human games
> We hypothesize that the difference in median length of human responses with human and SPC partners is caused by the template generation system in SPC. The template generation system asks questions that expect a yes or no answer without requesting other information from partners, allowing for shorter responses. A positive effect of SPC’s strategy is that humans require less effort overall to reach a similar success rate. We estimate human effort by the total number of human response words. The SPC strategy leads to shorter human responses and more turns, resulting in fewer human response words overall ($7.77$ turns $/ 2 * 6.95$ words $= 27.0 < 37.8 = 5.03$ turns $/ 2 * 15.06$ words, numbers from Tables 1 and 2).
>
> ## Difference in length of human responses with Imitate
> We hypothesize that the difference in human response length between SPC and the Imitate baseline was caused by the baseline requiring more effort and words from the partner in order to achieve success.
>
> ## Success rate for SPC in Fig 4 exceeding Table 1
> Thank you for catching this. The difference in the average success rate between the table and skill-based graph in Fig. 4 results from filtering out players who did not win any games for Fig. 4. This results in slightly higher success rates in Fig 4, which were greater than 70% opposed to 68.8% in Table 1. We will explain this in the next version of the paper.

---

### Meta-Review · Area_Chair_rPt2 · 2023-09-23

**Recommendation:** 4

**Metareview:**

The paper presents a novel method termed Symbolic Planning and Code-generation (SPC) that is applied to the OneCommon task. This task-oriented dialogue system hinges on a blend of neural (GPT-4 Large Language Model) and symbolic techniques to comprehend human interactions and formulate responses. SPC reads human utterances, generates code from them, and uses the code in a symbolic space for planning, subsequently reverting to natural language for responses. The evaluations indicate that SPC considerably surpasses the state-of-the-art method in human evaluations, especially when partnered with skilled humans. However, there are concerns regarding the clarity of the methodology, the system's broader applicability, and its novelty relative to prior research. Further elaboration on the process and addressing the cited limitations would bolster the paper's impact. After the rebuttal, it seems most concerns regarding soundness have been addressed, but reviewers are generally not very excited about this work. The AC feels that it would be a nice work to appear in the conference without all the additional materials and discussions.

---

### Decision · Program_Chairs · 2023-10-07

**Decision:**

Accept-Main

**Comment:**

The paper presents a novel method termed Symbolic Planning and Code-generation (SPC) that is applied to the OneCommon task. This task-oriented dialogue system hinges on a blend of neural (GPT-4 Large Language Model) and symbolic techniques to comprehend human interactions and formulate responses. SPC reads human utterances, generates code from them, and uses the code in a symbolic space for planning, subsequently reverting to natural language for responses. The evaluations indicate that SPC considerably surpasses the state-of-the-art method in human evaluations, especially when partnered with skilled humans. However, there are concerns regarding the clarity of the methodology, the system's broader applicability, and its novelty relative to prior research. Further elaboration on the process and addressing the cited limitations would bolster the paper's impact. After the rebuttal, it seems most concerns regarding soundness have been addressed, but reviewers are generally not very excited about this work. The AC feels that it would be a nice work to appear in the conference without all the additional materials and discussions.